# ADNP Syndrome: A Qualitative Assessment of Symptoms, Therapies, and Challenges

**DOI:** 10.3390/children10030593

**Published:** 2023-03-20

**Authors:** Jarrett Fastman, Alexander Kolevzon

**Affiliations:** 1Icahn School of Medicine at Mount Sinai, New York, NY 10029, USA; jarrett.fastman@mountsinai.org; 2Seaver Autism Center for Research and Treatment, Icahn School of Medicine at Mount Sinai, One Gustave L. Levy Place, P.O. Box 1230, New York, NY 10029, USA; 3Department of Psychiatry, Icahn School of Medicine at Mount Sinai, New York, NY 10029, USA; 4Department of Pediatrics, Icahn School of Medicine at Mount Sinai, New York, NY 10029, USA

**Keywords:** ADNP syndrome, autism, qualitative

## Abstract

ADNP syndrome is a neurodevelopmental disorder characterized by autism spectrum disorder (ASD), intellectual disability, sensory reactivity symptoms, facial dysmorphisms, and a wide variety of other physical and behavioral health manifestations. Research on ADNP syndrome has been limited, and there are currently no validated tools for assessing clinical outcomes in ADNP syndrome specifically. The goal of this qualitative study was to ascertain the symptoms of ADNP syndrome based on caregiver interviews, with the primary aim of identifying areas for clinical improvement that may inform the development of outcome measures specific to ADNP syndrome. Data collection consisted of loosely structured interviews with 10 caregivers of children with ADNP syndrome, representing 6 males and 4 females of ages 4 to 17 (M = 10.1; SD = 4.2). Interviews were conducted via phone between November 2020 and April 2021. The analysis of coded interview data identified three overarching themes: symptoms, therapies, and challenges. Each theme encompasses several distinct codes, which were individually addressed. Our results could ultimately be useful in educating clinicians about ADNP syndrome, selecting or designing refined outcome measures for clinical trials, and informing efforts to increase support for caregivers.

## 1. Introduction

ADNP syndrome, also known as Helsmoortel-Van der Aa syndrome, is a rare disorder in which sequence variants in the activity-dependent neuroprotective protein (ADNP) gene cause autism spectrum disorder (ASD), intellectual disability, sensory reactivity symptoms, facial dysmorphisms, and an array of other manifestations, including increased risk of seizures, cardiac defects, musculoskeletal issues, and gastrointestinal symptoms [1,2,3,4,5,6,7,8,9,10,11]. ADNP syndrome is among the most common single-gene causes of ASD, estimated to account for 0.17% of all cases [4]. 

Research on ADNP syndrome is complicated by both its rarity and heterogeneity of cognitive, behavioral, and physical symptoms across individuals. The qualitative characterization of individual cases is thus useful for identifying defining features of the disorder, and ultimately, for developing outcome measures for interventional studies. There are currently no validated tools for assessing clinical outcomes in ADNP syndrome specifically. Clinical outcome assessments typically used in ASD studies may not be well suited, given that many children with ADNP syndrome do not meet the full diagnostic criteria for ASD despite displaying many characteristic features [12,13]. As such, a greater understanding of the range, severity, and developmental course of symptoms is necessary to inform clinical endpoints and measurement in interventional studies. To date, only one small, open-label, single-dose clinical trial with low-dose ketamine has been performed in children with ADNP syndrome [14]. Information on patients’ experiences with medications and therapies may also be useful for selecting interventions for study. As these patients are complex and most have few-to-no words, input from caregivers is valuable for further characterizing the disorder. Yet, there have been no qualitative studies to examine the characteristics of ADNP syndrome based on in-depth information provided by caregivers. In addition, no studies have explored the unique practical challenges experienced by families affected by ADNP syndrome.

The goal of this qualitative study was to ascertain the symptoms of ADNP syndrome based on caregiver interviews. Further, we sought to establish specific changes that caregivers would consider to constitute meaningful improvement in those symptoms, with the aim of developing outcome measures for clinical trials. Information about each child’s medical history and past responses to interventions was collected, and taken together, may be useful for guiding clinicians unfamiliar with the disorder to provide appropriate referrals. In addition, many challenges faced by patients and caregivers were discussed, from obtaining the diagnosis to accessing treatment, which highlight the need for greater recognition and understanding of ADNP syndrome.

## 2. Materials and Methods

### 2.1. Population/Sample

The study population consisted of 10 caregivers of children with a diagnosis of ADNP syndrome. Participants were previously known to Seaver Autism Center for Research and Treatment at Icahn School of Medicine at Mount Sinai through enrollment in other studies or were recruited through ADNP Kids Research Foundation. ADNP Kids Research Foundation is available to advocate for families, increase awareness, and advance scientific understanding in order to develop treatments that improve the lives of children and families affected by ADNP syndrome (https://www.adnpfoundation.org/). Purposeful sampling was used to ensure that participants would be able to provide meaningful data. Participants included 9 mothers and 1 father, all of whom were primary caregivers to a child with ADNP syndrome. The children represented were 6 males and 4 females of ages 4 to 17 (M = 10.1; SD = 4.2).

### 2.2. Data Collection

Potential participants were contacted via email to schedule a one-on-one phone interview with the lead author (J.F.) between November 2020 and April 2021. At the time of the interview, verbal consent was obtained, and the interview was recorded for transcribing. Interviews ranged from 45 to 90 min in length and were concluded when the participant expressed that they had shared all that they wished. 

Data collection consisted of loosely structured interviews, and the following interview questions were used as prompts to continue the discussion when appropriate:

What were the first symptoms or behaviors that prompted concern about your child’s development?

In which areas would you most like to see improvement in your child?

What would you consider to be “meaningful improvement” in your child’s symptoms?

Which of your child’s symptoms are the most challenging to manage for you as their caregiver? How do you manage them?

What treatment options have you explored and what has been your experience?

Have you faced any barriers in obtaining treatment for your child?

What considerations are important to you when deciding whether to enroll your child in a clinical trial?

### 2.3. Data Analysis

Interviews were transcribed with identifying information redacted and manually coded with an inductive approach using open coding in MAXQDA software. Following the initial coding, a second pass of each interview was conducted to ensure the accuracy and consistency of codes across participants. Codes were then analyzed for frequency, proximity, and co-occurrence, and three code groups were created based on these characteristics. The codes within each code group were then examined for commonalities in order to define a unifying theme. Interview content was compared among participants in the context of each theme to identify commonalities and differences among cases.

## 3. Results

The analysis of interview data led to the identification of three overarching themes: symptoms, therapies, and challenges. While there is considerable overlap in their content, each theme describes a distinct element of the participants’ experiences and opinions. Our results are thus presented by theme, and further broken down into the codes that comprise each theme. As is standard in qualitative research, quotes from caregivers are also provided to illustrate themes.

### 3.1. Symptoms

Children with ADNP syndrome exhibit a wide range of medical and behavioral symptoms, many of which require complex management.


First symptoms and early development


“*She didn’t hit any of her developmental milestones at all. So it was pretty clear as early as like five months that there was something going on*.”

The majority of participants noted that their child did not meet developmental milestones within the first year of life. In several cases, dysmorphic features present from birth such as hypertelorism and low-set ears had raised immediate suspicion of a genetic disorder. Most children had been delayed in crawling and walking, and several had had delays in sitting upright, which was likely due to low muscle tone. Those who were verbal had been severely delayed in achieving speech. Lack of eye contact, minimal crying, and poor feeding had also been commonly noted in early infancy. All 10 caregivers interviewed noted that their children had exhibited premature tooth eruption.


Most severe or challenging symptoms 


“*Speech, cognition and sleep—those are the three things everybody talks about*.”

All participants cited problems with communication, cognition, and sleep as being among the most challenging symptom domains to manage on a daily basis. 

*Communication:* The symptom domain most commonly discussed by parents was speech and communication. Eight of ten children had few-to-no words, and these parents all identified their child’s inability to communicate their needs as a major source of stress for the entire family. Lack of verbal ability was seen by several parents as the trigger for aggressive and impulsive behaviors; these parents described their children as becoming frustrated and acting out when unable to communicate their needs. The parents of the two verbal children both expressed difficulty with social situations, as the children tended to say inappropriate or offensive things.

*Cognition*: All children exhibited cognitive impairment to varying degrees. Motor planning deficits, poor visuospatial skills, short attention span, and short-term memory deficits were common. Caregivers noted that cognitive symptoms were among the most debilitating aspects of ADNP syndrome, as they greatly contributed to the lack of independence in daily activities such as dressing and feeding.

*Sleep:* Most children had had difficulty with sleep since birth, most commonly in the form of frequent nighttime awakenings. This may have been a symptom of sleep apnea in at least some children (see “physical health problems”). The insomnia was severe in some cases, with one parent stating that they “can count on one hand the number of times [child] has slept through the night since she was born.” This lack of sleep did not seem to translate to daytime sleepiness, as most parents reported that their child was nevertheless highly active throughout the day. One family had seen improvement in sleep with a bowel regimen, and a second family had experienced improvement with a combination of guanfacine and clonazepam. However, the remaining eight caregivers interviewed stated they had not found any behavioral or pharmacological interventions to improve their child’s sleep.

*Motor skills:* All parents interviewed described a combination of gross motor, fine motor, and motor planning deficits. Most of the children were unable to feed themselves or hold a pencil due to poor grip and dexterity. All of the children exhibited motor planning deficits to some degree. While one was able to independently complete most activities of daily living, the majority had difficulty with any two-step motor process as well as movements that cross the midline of the body. Several children had also been diagnosed with low muscle tone, making them “wobbly” and prone to falls.

*Aggressive/impulsive and stereotypic behaviors:* All of the children exhibited aggressive, impulsive, and/or stereotypic behaviors to varying degrees. The most common aggressive behaviors were biting, hitting, hair-pulling, throwing objects, and self-injury. These behaviors were more prevalent in children with more severe language delay. Vocal stereotypies were also common. The frequency of maladaptive behaviors was noted to correlate with perceived frustration and anxiety.

*Sensory seeking:* Several parents expressed concern over sensory seeking behaviors. Oral fixation was the most common (“their hands are in their mouths 24/7”); one child had swallowed household items on numerous occasions, while another had recurrent episodes of cellulitis requiring hospitalization and intravenous antibiotics as a result of chewing on their fingers.

*High pain threshold:* The majority of parents stated that their child had an extremely high tolerance for pain. This often led to injuries going unnoticed by caregivers. One child had suffered multiple lacerations in his mouth due to a PROMPT speech therapy tool that had gone unnoticed for weeks, and several had had fractures and other injuries that had healed poorly due to having been initially overlooked.


Physical health problems



“*He was hospitalized half the year for the first three years of his life. We were living in the hospital*.”


All of the children had medical issues requiring specialist care. 

*Cardiovascular:* Two children were born with ventricular septal defects that had closed without intervention. One child had required multiple cardiac surgeries to repair “four holes in his heart;” the caretaker was not able to provide further details.

*Neurologic:* Three children had been diagnosed with cortical visual impairment and were legally blind. Of three children who had undergone brain magnetic resonance imaging, two revealed leukomalacia, and the third had thinning of the corpus callosum.

*Musculoskeletal:* The majority of children had issues with their feet, most commonly pronation and connective tissue tightness leading to an unstable gait. One child had required tendon lengthening surgery in their feet at age 5 and received monthly botox injections. Low muscle tone was also mentioned by three parents and is presumed to be the underlying cause of sleep apnea in the two children with that diagnosis.

*Gastrointestinal:* The majority of children in the study suffered from chronic constipation, and two had been diagnosed with gastroesophageal reflux disease.

*Other:* Two children had had frequent illness and recurrent infections, likely due to oral sensory seeking behaviors. Two children had hypothyroidism. One child was born with hypospadias requiring surgery, and another had required surgery for strabismus.


Clinical outcomes (meaningful improvement in symptoms)


“*If it could speed up her response time to things and help her focus, and maybe help with her short-term memory loss… I feel like that would make a real meaningful improvement in her life*.”

We asked participants to describe the clinical changes that would have most improved the quality of life for them and their child, and what they would have hoped to see if they enrolled their child in a clinical trial. All 10 participants stated that the most meaningful outcome would have been increased independence in activities such as feeding, dressing and toileting, as assisting with these activities required significant time and energy from caregivers. All participants also identified poor sleep as an important symptom to address. Several parents discussed improvement in verbal communication as particularly important to both reduce frustration and ensure that their child’s needs were being met. Cognitive issues such as motor planning deficits, short attention span, and poor short-term memory were also mentioned as meaningful clinical targets. Parents whose children were sensory-seeking also prioritized reduction in associated behaviors, particularly oral fixation.

In regards to participating in clinical trials, all interviewees stated that they would have been willing to enroll their child in any study that might have provided benefit. Several parents expressed concerns about access to clinical trials and disappointment if participation required extensive travel.

### 3.2. Therapies

Participants had made use of a wide variety of medical and behavioral therapies. Most described having seen improvement in symptoms with intensive therapy, but nearly all had had difficulty in accessing an adequate level of care.


Effective therapies


“*She has a one-to-one behavioral therapist with her all day and gets PT [physical therapy], OT [occupational therapy], speech and PROMPT [speech] therapy all in school, plus 15 h per week of a behavioral therapist at home and an at-home oral motor therapist… I can’t tell you how many things we’ve tried*.”

Study participants had found varied success with a broad range of therapies. The most commonly utilized and overall most successful intervention discussed was applied behavioral analysis (ABA) therapy. ABA is a technique used in children with developmental disabilities that focuses on teaching skills in specific domains of functioning using traditional principles of positive reinforcement. All parents had utilized ABA, with the greatest benefits being seen in children who were able to receive therapy both at school and at home. While some parents had found ABA to be minimally effective, others had seen marked reductions in problem behaviors with consistent therapy. Physical and occupational therapy were considered important and generally effective by all parents interviewed, though again, the degree of efficacy was highly variable. For several children, speech therapy (including PROMPT therapy) had improved communication, though a subset of children had not responded to speech therapy. One child had exhibited significant improvement in symptoms with music therapy and horseback riding (hippotherapy); another child had also benefited from hippotherapy.

The majority of participants had trialed at least one medication for their child’s symptoms, but few had found success with pharmacologic treatments. As noted, one child had had improvement in problem behaviors and sleep with guanfacine and clonazepam. Another child had exhibited significantly improved attention and impulse control with methylphenidate but discontinued the medication after developing motor tics. Several families had been able to effectively manage chronic constipation with laxatives and diet alterations.


Ineffective therapies


“*I don’t think you’re going to solve any of his problems with medications.*”

All participants had tried multiple therapies that had not provided significant benefit to their child. While ABA therapy had been effective for most children, one parent believed that it had worsened their child’s symptoms and caused behavioral regression to the point where they “didn’t even recognize him anymore.” For children with cortical visual impairment and/or difficulty focusing their attention, speech therapy was not seen as viable or effective. Several parents reported having tried various stimulant and antidepressant medications for behavioral control, but these had most often produced adverse effects without clear benefit. The majority of parents had trialed multiple interventions for sleep, including diet changes, light therapy, melatonin, benzodiazepines, and magnesium, all with minimal success.


Need for early diagnosis and intervention 


“*He got the basic therapies, and it wasn’t until after his diagnosis that we started asking for aggressive therapies. And then come to find out, this is a neuroplasticity-blocking gene… but a lot of the younger kids are getting more intensive, almost rehab-level therapy and they’re crushing it. They’re saying the days of the week and singing songs*.”

The importance of early diagnosis and intervention for improved long-term outcomes in ADNP syndrome was repeatedly reported throughout the interviews. Most participants described a strong belief based on personal experience that the earlier in an affected child’s life treatment is begun, the less delayed the developmental trajectory is. One parent stated that their child had one of the most severe cases known and believed this to be due to a lack of therapy in early childhood. Another parent attributed their child’s relatively mild symptoms to early intervention, stating that “as disabled as she is, when I see the other kids with ADNP [syndrome] I know that the reason she’s as high functioning as she is, is the therapy she got at a young age.” Two other parents whose children were comparatively high-functioning attributed this to obtaining early diagnosis and treatment. 


Frequency and intensiveness


“*What ADNP kids need is frequency, frequency, frequency. They need to keep it up. If you don’t do it, you lose it—and you need to do a lot of it*.”

Most participants felt that a minimum of several hours per day of one-on-one therapy should be the standard for children with ADNP syndrome. While the parent of one child had noted improvement in symptoms with as little as 2 hours of therapy daily, most children received at least 4 hours, and some had only seen benefit with 8 or more hours daily. The amount of therapy received by the children in our study ranged from 2 hours on weekdays to 12 hours daily including weekends. Several parents mentioned the importance of consistency and believed that their child only benefited when receiving therapy 7 days per week. A combination of at-school and at-home behavioral therapies was seen as most effective. One participant reported that in the past, their child had failed to make any measurable progress even when receiving daily therapy; however, they had noted major improvements in a number of symptom domains after being approved for an at-home behavioral therapist eight hours per day. Another caregiver stated that getting a full-day one-on-one therapist for their child at school “has made all the difference in the world,” and that being able to supplement therapy at school with at-home therapy had accelerated their child’s progress. The COVID-19 pandemic was mentioned by several parents as a major barrier to receiving adequate care at the time of interview, and two reported having seen significant regression in their child’s behaviors as a result. Some parents also expressed that even before the COVID-19 pandemic, they often worried that they could lose access to services at any time for financial or insurance-related reasons, and that a lack of consistency in therapy could lead to regression.

### 3.3. Challenges

There were several challenging aspects of managing ADNP syndrome discussed by participants that were distinct from the other identified themes and highlight challenges faced by caregivers that are unique to parenting a child with a rare genetic disorder. 


Difficulty accessing care


“*We needed to have therapy in the house, which is really hard to find, almost impossible to find in-network for a child… and so he’s getting very little services*.”

The majority of participants identified obtaining adequate care for their children as one of the most consistently difficult things they had encountered. Children with ADNP syndrome likely require consistent, intensive physical and occupational therapy at a level that is challenging to find and to get reimbursed for by insurance. While most participants reported that they were currently satisfied with the care their child received, nearly all shared several examples of times when their child had been forced to go for extended periods with little or no daily therapy. The lack of adequate disability accommodation and therapy at public schools in particular was identified as a major problem. For several parents, geography posed an issue, as there were very few behavioral therapists in their respective areas, and waitlists could be years long. 

All interviewees brought up insurance coverage and described having dedicated significant time and energy to attempts to have services reimbursed. While some families were able to pay out of pocket for intensive therapy when not covered, others were severely limited by their insurance. One parent expressed frustration at not being able to access at-home therapy, which was the only viable option for their family as their child had severe separation anxiety.


Obtaining an autism diagnosis


“*The autism diagnosis was really crucial for us. They wouldn’t have covered anything if we didn’t have that.” “I can’t tell you how many families, no matter where they’re from… they all try to get autism diagnoses for their kids when they’re little, but they don’t meet the criteria because they’re too social*.”

Nine of ten participants indicated obtaining a diagnosis of ASD as a critically important step in obtaining adequate care for their child. While in the majority of cases, the diagnosis of ASD preceded the diagnosis of ADNP syndrome, most participants expressed that they did not believe that their child had ASD but that they would have had significantly less access to care without the diagnosis (*“She has ADNP syndrome, she doesn’t have autism. But that diagnosis has been the only thing that has gotten us any reimbursement with insurance.”*). However, the majority of parents had undergone a lengthy process of fighting for the ASD diagnosis, and several described the frustration of waiting months or years knowing that their child was not receiving appropriate therapies that could potentially alter their developmental trajectory. The most common obstacles to obtaining an ASD diagnosis were long wait times for clinical evaluations and patients not meeting diagnostic criteria due to being too socially high-functioning.


Need for advocacy


“*We need to advocate. We need people to stand up and take stances on behalf of these children…*”

Parents of children with ADNP syndrome must often advocate for school accommodation and medical services, and on a larger scale, for greater recognition of the syndrome. All the participants spoke extensively of the need to advocate for their children in a variety of contexts, often due to a lack of knowledge and understanding about ADNP syndrome. The majority of parents interviewed had spent a significant amount of time and energy negotiating with their insurance companies to get services covered; one parent likened this to having a second full-time job. All participants had had difficulty arranging for adequate at-school accommodation and therapy, and three had sued their school districts to varying degrees of success. Several parents also described feeling a responsibility to advocate for research on ADNP syndrome and to “lead the scientific effort to understand it.”


Uncertainty about diagnosis and prognosis


“*We didn’t have our answer until she was almost 7... the second whole [exome] sequencing we did came back with ADNP. The first one, ADNP wasn’t discovered yet, so we got no hits.” “It was really scary and hard to kind of wrap our minds around. What does this mean for our child? And what type of life can we lead with this syndrome that nobody really knows anything about?*”

Participants described the impact of uncertainty in caring for a child with ADNP syndrome, from the initial onset of symptoms to diagnosis and treatment. Given the rarity of the disorder, the process of obtaining a diagnosis is often lengthy and difficult. Even following the diagnosis, caregivers may be forced to make decisions about treatment with limited information. The majority of families had gone several years without the genetic diagnosis; one child had not received a diagnosis until age 16. While receiving an ADNP syndrome diagnosis was relieving to parents, in most cases, it created a great deal of stress and uncertainty given that there “was no roadmap.” Parents described feeling worried about their child’s prognosis and their ability to ensure that their child’s needs were met into adulthood.

## 4. Discussion

Our results describe a broad range of symptomatology in ADNP syndrome and highlight practical issues experienced by children with ADNP syndrome and their families. The lack of verbal communication, cognitive delay, and sleep disturbance were the most common and debilitating symptoms identified by caregivers. In terms of physical health issues, musculoskeletal and gastrointestinal issues were most common, with visual and cardiac defects also being present in some children. Awareness of these highly prevalent behavioral and physical health problems may aid clinicians in the evaluation of children with newly diagnosed ADNP syndrome. It should be noted that while only three patients in our study had undergone neuroimaging, all had pertinent findings, and the imaging of additional patients may reveal a characteristic pattern of brain abnormalities worthy of monitoring. Phenotypic features such as hypertelorism, low-set ears, and premature tooth eruption were common and are consistent with those reported in other studies [3]. Given these results, children diagnosed with ADNP syndrome should at a minimum be considered for specialist cardiac, gastroenterological, and neurological evaluations. 

The need for early diagnosis and intervention to improve long-term outcomes was a common sentiment amongst all interviewees. The lowest functioning children in our study were those who had not received any treatment in early childhood, and conversely, the children who had had access to intensive therapy from an early age had greater verbal ability and fewer problem behaviors. Our preliminary results suggest that generally speaking, behavioral interventions are more effective than medications in reducing problem behaviors. ABA therapy was the most commonly used and was reported by nearly all interviewees to be at least somewhat effective, with those whose child received ABA both at home and at school reporting the greatest benefit. PROMPT therapy has been described to lead to improvement in verbal ability in a subset of children but has not been effective for others. All interviewees shared the belief that the efficacy of these interventions is directly related to their age of initiation, frequency, and intensity, and several noted that progress was rapidly lost whenever therapy was reduced or discontinued. The most profound improvements were reported by parents whose children received at least four hours of one-on-one therapy daily.

Several parents mentioned feeling uncertain about what kind of medical care their child might need upon receiving an ADNP syndrome diagnosis. In terms of specialist referrals, children with ADNP syndrome should likely undergo cardiac, gastroenterological, ophthalmic, and neurologic evaluations based on the symptoms described by our participants. As connective tissue tightness appears to be common, some children need orthopedic interventions. Given the abnormal findings of MRI of the two children in our study on whom it had been performed, neuroimaging may be useful in children with ADNP syndrome to establish a baseline and for monitoring as needed. In addition, children with ADNP syndrome should be closely monitored for signs of injury or infection given the prevalence of oral sensory seeking behaviors and high pain tolerance.

For future clinical trials, outcome measures should include assessments of activities of daily living skills, sleep, and sensory reactivity, as these symptoms may be tractable even in short-term clinical trials. Further, the measurement of communication and performance in cognitive domains such as motor planning and memory should be explored if feasible. Deficits in these domains were both prevalent and frequently identified as meaningful clinical targets by interviewees. Despite the fact that many children with ADNP syndrome do not meet the full criteria for ASD, clinical tools validated for assessing ASD symptoms domains may also be useful to include in clinical trials. Our results may be used to inform the adaptation of existing clinical tools for ASD to address the unique features of ADNP syndrome.

The most significant and pervasive practical issue faced by families in our study was the ability to access care and establish supportive services. Some families had to relocate in order to access services due to lack of availability or prohibitively long waitlists for clinical evaluations. Several parents expressed frustration over their inability to enroll in clinical trials due to the requirement for travel. Multi-site clinical trials with collaboration across institutions in different geographic locations may increase accessibility of research participation for families. Additionally, the need for advocacy with greater public awareness of ADNP syndrome is crucial to prompting government agencies and insurance companies to include coverage of services with or without the need for an ASD diagnosis.

Our results could ultimately be useful for educating clinicians about ADNP syndrome, selecting or designing refined outcome measures for clinical trials, and informing efforts to increase support for caregivers. Future studies should aim to further characterize the range of ADNP phenotypes in larger samples using prospective and longitudinal study designs. Practice parameter guidelines are needed to aid clinicians and ensure the comprehensive evaluation, monitoring, and treatment of individuals with ANDP syndrome.

## Data Availability

The data presented in this study are available upon request from the corresponding author.

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
