# Peer review of "ADNP Syndrome: A Qualitative Assessment of Symptoms, Therapies, and Challenges"

_children, 2023, doi:10.3390/children10030593_

Round 1

Reviewer 1 Report

This is a very interesting paper in a very under researched area. Thank you for the work. There are only two quite minor issues I wish to pint out

1/ p. 2 - line 70 parents and primary caregivers - I just want to be clear that all participants were in both roles.

2/ p. 4 line 128 - this sentence needs a minor rewrite - as presently written that the aggressive and violent behaviors were by the parents. I think it should be - lack of verbal ability was seen by the parents as triggers for

Reviewer 2 Report

This is a very important article that I expect will become a classic in ADNP Syndrome research. Clearly, this will be a highly cited article. This is the article that parents are likely to bring to medical professionals when searching for answers regarding their children. Therefore, it is important to mine the data for as much as possible.

A sample of 10 for this kind of research is fine. The authors need to clarify that all 10 are parents of the children with ADNP syndrome. Line # 65 refers to "caregivers." This word can mean different things to different people. "Caregivers" can be parents, other relatives, neighbors, or various kinds of hired professional or nonprofessional help. At various points in the article there is reference to primary caregivers, participants, and interviewees. If all 10 are parents, I recommend sticking with the "parent" designation. The Abstract needs to replace "caregiver" with parent also. If I am reading this incorrectly, then the authors need to be specific about others besides the parents who provide care.

What can be said about the families? Of the 10, are there a mix of mothers and fathers? If so, that should be stated. If all 10 are mothers that should be stated also. Can the authors report if there are other children in these families?

The authors needs to be more explicit in introducing the sentences quoted from parents. This is standard in qualitative research. For example, "One mother of a boy stated:...."

The authors should provide a definition of ABA--a sentence or two around line #221.

It is difficult to get a sense of how much professional intervention children are receiving. It seems to be an average of about 4 hours a day. At least one child is receiving 8 hours a day. Can the authors be more explicit than parents wanting "several hours a day?" (line #s 271-272). What more can be said about this?

In line # 288 "which" should be replaced by "that."

Readers are struck by the extent of disability of these children--the type and amount of care they need and the severity of the  behavior problems the children experience and that families must cope with. Parents must live in a state of being overwhelmed and must have difficulty meeting their own needs for relaxation and sleep. If there are other children in the home, I expect parents may feel guilt that the child with ADNP must receive much more of their attention. Can the authors say anything about parental coping mechanisms?

Three of the 10 parents have filed lawsuits to ensure their children receive services. This really speaks to the desperation parents must be feeling. What more can the authors say about these lawsuits. Are they pending? Resolved? Was this a successful avenue? A huge expense for a lawyer?

As the parent of a severely disabled child, I do not like the term "caregiver burden." Parents do not want to think of their children as a burden. I suggest replacing the term throughout with a term already used in the article, "challenges."

Are the authors willing to include something about the ADNP Kids Research Foundation? A short statement of purpose and services would be helpful. The Foundation offers Facebook group support for parents, a much-needed resource.

This article will be read by a variety of professionals outside of the medical field since it is appearing in this specific journal. Human service workers, social workers, educators, and others will find this article of great interest. Thus I suggest a more comprehensive conclusion that reiterates the critical point that the sooner a child receives services, the better the outcomes. Also, reiterate the need for neuroimaging  and the specialty evaluations from cardiology, gastroenterology, and neurology.

A preferred title for the authors to consider:

ADNP Syndrome: A Qualitative Assessment of Symptoms, Therapies, and Challenges

Overall, an excellent effort!

Author Response

Please see the attachment. Thank you for your comments and suggestions!
